# Design and Development of an Air–Land Amphibious Inspection Drone for Fusion Reactor

Guodong Qin [1], Youzhi Xu [2], Wei He [2], Qian Qi [2], Lei Zheng [1,2], Haimin Hu [3], Yong Cheng [1], Congju Zuo [1,3,*], Deyang Zhang [4,*] and Aihong Ji [2,5,*]

1   Institute of Plasma Physics, Chinese Academy of Sciences, Hefei 230031, China; gdqin@ipp.ac.cn (G.Q.); zhenglei@ipp.ac.cn (L.Z.); chengyong@ipp.ac.cn (Y.C.)
2   Lab of Locomotion Bioinspiration and Intelligent Robots, College of Mechanical and Electrical Engineering, Nanjing University of Aeronautics & Astronautics, Nanjing 210016, China; xuyouzhi@nuaa.edu.cn (Y.X.); hw15vc@163.com (W.H.); qiqian@nuaa.edu.cn (Q.Q.)
3   Department of Information Engineering, PLA Army Academy of Artillery and Air Defense, Hefei 230031, China; hmhu@163.com
4   Zhengzhou Campus, PLA Army Academy of Artillery and Air Defense, Zhengzhou 450052, China
5   State Key Laboratory of Mechanics and Control for Aerospace Structures, Nanjing University of Aeronautics and Astronautics, Nanjing 210016, China
*   Correspondence: judy@mail.ustc.edu.cn (C.Z.); zhangdeyang@alu.hit.edu.cn (D.Z.); meeahji@nuaa.edu.cn (A.J.)

**Abstract:** This paper proposes a design method for a miniature air–land amphibious inspection drone (AAID) to be used in the latest compact fusion reactor discharge gap observation mission. Utilizing the amphibious function, the AAID realizes the function of crawling transportation in the narrow maintenance channel and flying observation inside the fusion reactor. To realize miniaturization, the mobile platform adopts the bionic cockroach wheel-legged system to improve the obstacle-crossing ability. The flight platform adopts an integrated rotor structure with frame and control to reduce the overall weight of the AAID. Based on the AAID dynamic model and the optimal control method, the control strategies under flight mode, hover mode and fly–crawl transition are designed, respectively. Finally, the prototype of the AAID is established, and the crawling, hovering, and fly–crawling transition control experiments are carried out, respectively. The test results show that the maximum crawling inclination of the AAID is more than $20°$. The roll angle, pitch angle, and yaw angle deviation of the AAID during hovering are all less than $2°$. The landing success rate of the AAID during the fly–crawl transition phase also exceeded 77%, proving the effectiveness of the structural design and dynamic control strategy.

**Keywords:** fusion reactor; air–land amphibious; inspection drone; wheel-legged system; optimal control

## 1. Introduction

Inside the fusion reactor, there are thermal loads, electromagnetic forces, neutron radiation, etc., causing cracks, fissures, and other defects in the core components. It is necessary to carry out regular inspections and maintenance of the core components with the help of remote handling systems to ensure efficient and stable operation of the device [1,2]. Existing vacuum chamber inspection equipment is dominated by multi-joint robotic arms, such as the in-vessel viewing system (IVVS) of the International Thermonuclear Experimental Reactor (ITER) and EAST Articulated Maintenance Arm (EAMA) of China [3,4]. They all have the shortcomings of huge volume, expensive cost, slow-moving speed, low inspection efficiency, etc., which will be unable to cope with the inspection tasks of the bigger and more complex fusion reactor devices in the future. A miniaturized and multi-functional inspection drone system can be designed for fusion reactor inspection and maintenance tasks, as shown in Figure 1. It can detect defects such as target plate detachment and target plate damage in the first wall of the fusion reactor. The efficiency of the fusion

reactor inspection and maintenance tasks can be improved through the autonomous control of the whole process of transportation, takeoff, inspection, landing, and recovery of the inspection drone [5–7]. The narrow working space inside the fusion reactor makes the structural design and transportation of the drone more difficult. Currently, under CFETR maintenance requirements, the drone system shares a transport channel with the multi-jointed endoscopic system [8]. In standby mode, it needs to be stored in the transport channel; and in working mode, it must autonomously enter the vacuum chamber from the transport channel to complete the take-off inspection, so the drone system must have amphibious movement capability. Through research, it has beenfound that the air–land amphibious inspection drone (AAID) can enact fly–crawl amphibious conversion, and the integrated structural design can meet the transportation, take-off, and landing problems of the inspection drone in the maintenance port [9–11].

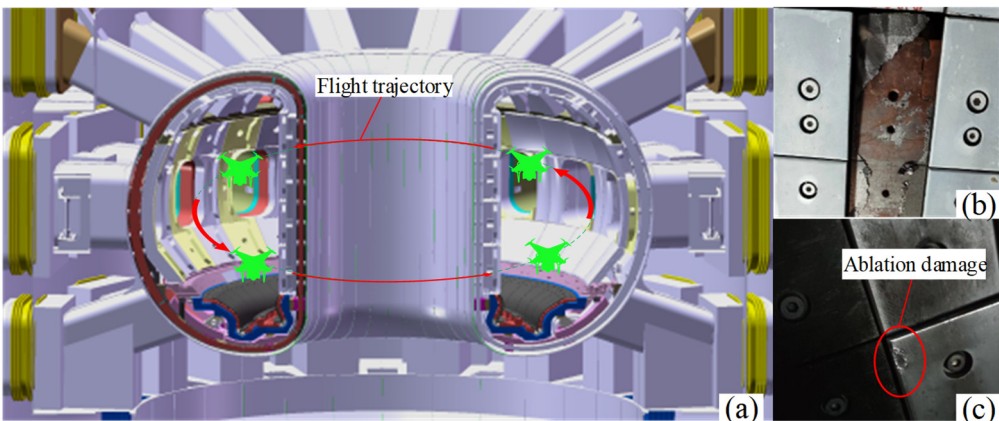

**Figure 1.** Inspection drone system for fusion reactor environments: (**a**) working trajectory (follow red arrows); (**b**) first wall target plate detachment; (**c**) target plate ablation damage (inside red circle).

Inspection drones can be categorized as airborne, ground, and waterborne, according to the takeoff method [12–14]. Air–land amphibious drones are equipped with ground-crawling and airborne flight capabilities to perform ground and airborne tasks, respectively [15,16]. This multi-functional vehicle, also named "flying car", can fly like an airplane and drive on the road like a car [17]. Air–land amphibious drones can move based on the negative pressure generated by the rotor blades, reducing the complexity of the drive system. It is also possible to design an independent drive system to realize the movement of the drone on land using tracks, wheels, legs, etc. [18]. For amphibious drone mobile platforms, Bachmann et al. proposed a miniature air–land drone platform that can fly and walk over rugged landscapes [19,20]. Currently, many factors are hindering the application of miniature mobile platforms for drones, such as unstructured environments (stairs, rubble, undulating terrain, etc.) [21–23]. Secondly, the miniaturization of the power supply also makes it difficult to keep up with other key devices (e.g., actuators, sensors, and calculators), which creates difficulties in the design of mobile platforms [24,25]. For the design of drones integrated with mobile platforms, [26] proposed a three-rotor mechatronic drone design method, in which three waterproof motors were attached to an isosceles-triangular omnidirectional wheel to enable movement over water after landing. It could also be equipped with omnidirectional mechanical wheels for air–land amphibious movement [27]. Baker et al. realized the conversion control of a drone from a miniature tracked platform to an airborne quadrotor through a hybrid design study [28]. Through the bionic design of a vampire bat, Daler et al. proposed a DALER amphibious drone design method. It consists of a flying wing with an adaptive morphology that allows the robot both to fly over long distances and walk in the target environment [29]. To integrate the advantages of drones and mobile platforms, a typical design concept is to directly integrate and design the propellers as mobile platform drive wheels to realize amphibious movement capability [30–32]. In addition, designing air–land separation amphibious drone systems

through cooperative control is also an effective solution [33–35]. The ground platform in this method is mainly used as a support device for drones. It can fly in 3D space to share airborne and land information and can easily avoid terrain changes and obstacles; thus, it is promising in terms of collaborative tasks [36–38]. However, to ensure accurate alignment of the drone with the mobile platform, it is necessary to introduce active and passive alignment mechanisms, increasing the technical and design difficulties [39–41]. At present, research on the AAID is still in its primary stage, and it is of great significance for the safe and smooth operation of fusion reactors to carry out research on small-scale air–land amphibious drone systems for the actual scenario of fusion reactor discharge gap observation.

In this paper, an air–land amphibious inspection drone (AAID) design method is proposed for the fusion reactor inspection tasks. First, a miniature hexapod wheel-legged mobile platform is proposed to realize ground crawling through a cockroach motion bionic design. Then, an amphibious movement capability is realized by combining it with a quadrotor platform. By integrating the flying and crawling amphibious function, our method addresses the need for crawling transportation in a narrow maintenance port and flying observation inside the fusion reactor. Based on the AAID dynamic model, the flight and fly–crawl transition control laws are designed to realize air–land coordination. Finally, the AAID prototype is produced for prototype testing in an indoor environment. This paper is divided into five sections: Section 2 introduces the design and modeling methods for air–land amphibious drones; Section 3 describes the drone flight and fly–crawl transition control strategies; Section 4 establishes the experimental platform and conducts experiments on the flight, crawl, and fly–crawl transition capabilities of the drone; and finally, Section 5 gives the conclusion of this paper.

## 2. Design and Modeling of the AAID

### 2.1. AAID Bionic Design

Based on the internal environment of China's latest compact fusion reactor and the inspection and maintenance requirements, the AAID design parameters are established as shown in Table 1. They mainly include the magnetic field parameters inside the vacuum chamber, the pressure parameters, the transport mode observation accuracy, and other technical indexes. The AAID will be innovatively designed around the above parameters.

**Table 1.** Fusion reactor inspection drone design requirements.

| Items | Parameters |
|---|---|
| Magnetic field during inspection | 0.05 T |
| Magnetic field during storage | 0 |
| Pressure during the storage | $\leq 10^{-5}$ Pa |
| Pressure during inspection | Atmospheric pressure, dry air, or nitrogen |
| Drone size requirements | $\leq 150$ mm $\times$ 150 mm $\times$ 150 mm |
| Storage pipeline transport mode | Auto-crawl |
| Inspection process movement mode | Auto-flight |
| 0.5–4 m normal observation accuracy | $\leq 3$ mm |

The amphibious movement of the AAID requires a miniature crawling device to drive it. It is observed that the cockroach adopts an alternating triangular gait for crawling movement, as shown in Figure 2a. Based on the motion gait of cockroaches, a design method of a wheel-legged crawling system with a wide middle and two narrow ends is proposed as shown in Figure 2b. The wheel-leg consists of the center wheel body and curved leg pieces. Wheel-legs at both ends of the same side of the wheel axle are mounted in different ways, and the main design parameters are shown in Table 2. The advantages of the wheel leg design include the following: (1) the contact point between the wheel leg and the ground is closer to the wheel axle, which reduces the load on the motor; (2) when the roughness of the contact surface is not enough, the anti-skid performance of the curved

structure is better; and (3) the curved wheel leg has better obstacle-crossing ability when encountering complex terrain. The force analysis of the crawling system is shown in Figure 2b,c, which shows that when the system advances, due to the unevenness of the contact surface, the friction force $F_f$ received by the wheel-leg alternation is not parallel to the motion direction but is at a certain angle ($\alpha$) with the motion direction. For creeping insects, the $\alpha$ angle allows for greater stability of the locomotion system and greater grip. When the wheel legs meet an obstacle, it can prevent tipping to some extent. The torque of the motor at the starting instant of the crawling system is

$$T_0 = F_f r_g cos\alpha. \tag{1}$$

In the phase of uniform-speed movement of the crawler system, we assume that the angle of rotation of the leg of the crawler system is $\Phi$, and its corresponding torque is

$$T_t = F_g r_g cos\Phi + F_f r_g cos\alpha sin\Phi. \tag{2}$$

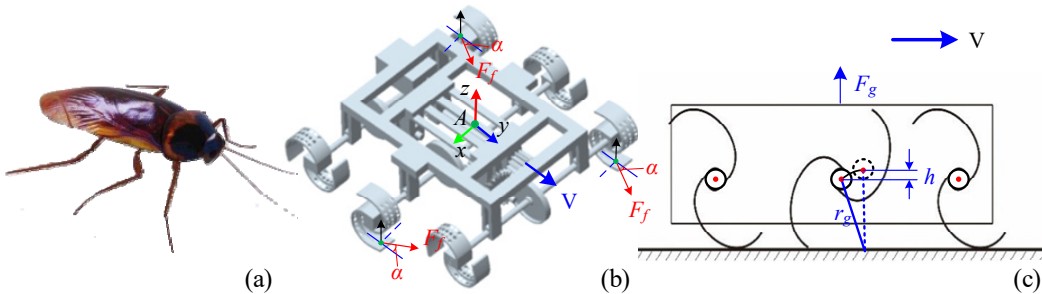

**Figure 2.** Wheel-legged crawling system design: (**a**) cockroach crawling gait; (**b**) principle of wheel-leg structure design; (**c**) force analysis of wheel-legged crawling system.

**Table 2.** Parameter of wheel-legged crawling system.

| Items | Parameters |
|---|---|
| Wheel-leg mass | $\leq$25 g |
| Chassis size | 50 mm $\times$ 50mm $\times$ 15 mm |
| Wheel-leg size | 20 mm $\times$ 15mm $\times$ 6 mm |
| Crawling speed | >1.5 m·s$^{-1}$ |
| Crawling ability | $\geq$20° |
| Overrun height | $\geq$10 mm |

The drive system of the AAID crawler uses a single motor to drive the three axles to rotate synchronously. The DC motor with a rated torque of 0.512 mN·m is selected to drive a two-stage gear reduction mechanism consisting of a main shaft gear, a crown gear, and a follower gear with a gear ratio of $i = 8.4$. Assuming that the force on the ground applied to the crawling system is $F_g = G = 0.25$ N, the linear length of the wheel legs $r_g = 10$ mm, and the range of undulation of the center of gravity is approximately $h = 1$ mm. The total torque required for the three wheel legs at this point is

$$T_d = G\sqrt{r_g^2 - h^2} = 2.44 \text{ mN·m}. \tag{3}$$

From Equation (2), the maximum torque output from the motor is calculated as

$$T_m = \frac{9550P}{n}ii = 4.29\text{mN·m}. \tag{4}$$

From Equations (3) and (4), it can be seen that $T_m > T_d$, which meets the drive design requirements. The flight system of the AAID is designed using a quadrotor structure with an integrated design of rack and flight control, as shown in Figure 3a. The size is

100 mm × 100 mm, which has a certain load capacity and can also meet the miniature requirement. By integrating the weight and size of the bionic wheel-leg structure and the miniature quadrotor, the design model of the AAID system can be obtained as shown in Figure 3b.

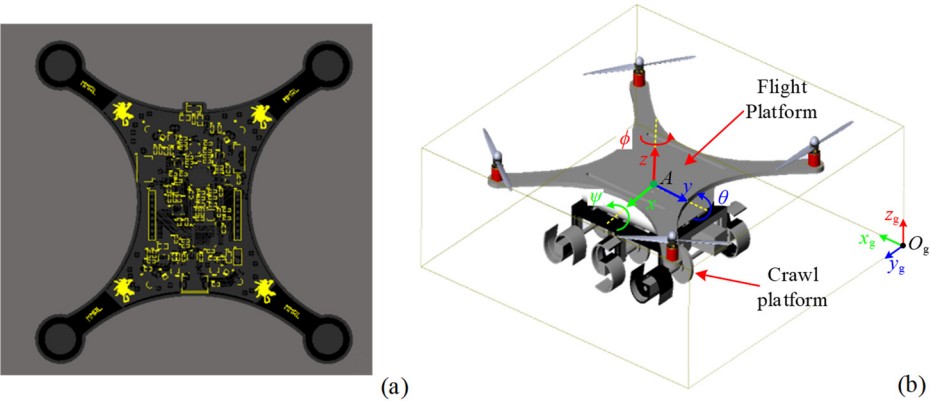

(a)                (b)

**Figure 3.** The integrated design of the AAID flight and crawling system: (**a**) AAID control board design; (**b**) AAID entire model and coordinate system.

### 2.2. Dynamic Modeling

As shown in Figure 3b, the angle of rotation around the *X*-axis is defined as the pitch angle $\psi$, the angle of rotation around the *Y*-axis is defined as the roll angle $\theta$, and the angle of rotation around the *Z*-axis is defined as the yaw angle $\phi$ in the AAID airframe coordinate system. The transformation matrix from the airframe coordinate system to the ground coordinate system is

$$R = \begin{bmatrix} c\theta c\psi & c\psi s\theta s\phi - s\psi c\phi & c\psi s\theta s\phi + s\phi s\psi \\ c\theta s\psi & s\phi s\theta s\psi + c\phi c\psi & s\psi s\theta c\phi - s\phi c\psi \\ -s\theta & c\theta s\phi & c\theta c\phi \end{bmatrix}, \tag{5}$$

where *c* denotes *cos*, and *s* denotes *sin*. The main forces on the robot in flight mode are the propeller pull, air resistance, torque from rotor rotation, and gravity. According to Newton's second law of motion, we can obtain

$$\begin{cases} \vec{F} = m\dfrac{d\vec{V}}{dt} \\ \vec{M} = \dfrac{d\vec{H}}{dt}, \end{cases} \tag{6}$$

where $\vec{F}$ denotes the sum of external forces on the AAID, $\vec{V}$ denotes the velocity of the center of mass, $\vec{M}$ denotes the combined external torque concerning a certain axis, *m* denotes the mass of the AAID, and $\vec{H}$ denotes the momentum torque of the AAID. The total lift generated by the AAID is *T*, and the lift of each rotor is $T_i$. In the flight mode, the lift applied to the AAID is upward perpendicular to the fuselage plane. Then, the lift $F_B$ in the fuselage coordinate system and the lift $F_E$ in the ground coordinate system can be expressed, respectively, as

$$F_B = \begin{bmatrix} 0 & 0 & T \end{bmatrix}^T; \tag{7}$$

$$F_E = RF_B = T \begin{bmatrix} c\psi s\theta c\phi + s\phi s\psi \\ s\psi s\theta c\phi - s\phi c\psi \\ c\theta c\phi \end{bmatrix}. \tag{8}$$

Ignoring the flight drag, the displacement of the AAID in the reference coordinate system is

$$
\begin{bmatrix} \ddot{x} \\ \ddot{y} \\ \ddot{z} \end{bmatrix} = \frac{F_E}{m} - g \begin{bmatrix} 0 \\ 0 \\ 1 \end{bmatrix} = \begin{bmatrix} T\frac{(c\psi s\theta c\phi + s\phi s\psi)}{m} \\ T\frac{(s\psi s\theta c\phi - s\phi c\psi)}{m} \\ T\frac{c\theta c\phi}{m} - g \end{bmatrix}.
\tag{9}
$$

The aerodynamic rotor lift versus rotational speed is given by

$$
T = b\sum_{i=1}^{4} w_i^2,
\tag{10}
$$

where $b$ is the rotor lift coefficient and $w_i$ ($i = 1, 2, 3, 4$) is the rotational speed of the propeller.

According to the Euler equation of the rigid body, the dynamic equation for the rotation process of the drone can be obtained as

$$
\begin{cases} M_x = J_x\ddot{\psi} + (J_z - J_y)\dot{\theta}\dot{\phi} = \frac{\sqrt{2}}{2}L(T_1 + T_2 - T_3 - T_4) + I_w\dot{\theta}(w_1 - w_2 + w_3 - w_4) \\ M_y = J_y\ddot{\theta} + (J_x - J_z)\dot{\psi}\dot{\phi} = \frac{\sqrt{2}}{2}L(T_1 - T_2 - T_3 + T_4) + I_w\dot{\psi}(w_1 - w_2 + w_3 - w_4) \\ M_z = J_z\ddot{\phi} + (J_y - J_x)\dot{\psi}\dot{\theta} = d(-w_1^2 + w_2^2 - w_3^2 + w_4^2), \end{cases}
\tag{11}
$$

where $\psi$, $\theta$, and $\phi$ denote the roll, pitch, and yaw angles in the airframe coordinate system, $J$ denotes the inertia matrix, $I_w$ denotes the inertia of rotation, $M$ denotes the combined torque applied to the airframe, and $M_x$, $M_y$, and $M_z$ denote the torques applied to the airframe in the three directions. From Equation (11),

$$
\begin{cases} U_1 = T_1 + T_2 + T_3 + T_4 = b(w_1^2 + w_2^2 + w_3^2 + w_4^2) \\ U_2 = T_1 + T_2 - T_3 - T_4 = b(w_1^2 + w_2^2 - w_3^2 - w_4^2) \\ U_3 = T_1 - T_2 - T_3 + T_4 = b(w_1^2 - w_2^2 - w_3^2 + w_4^2) \\ U_4 = -T_1 + T_2 - T_3 + T_4 = d(-w_1^2 + w_2^2 - w_3^2 + w_4^2), \end{cases}
\tag{12}
$$

where $U_1, U_2, U_3$, and $U_4$ denote the control quantities for the four channels of vertical, roll, pitch, and yaw, respectively, $d$ denotes the anti-torque coefficient, and $L$ denotes the length of the quadrotor arm in the AAID flight module. Assuming the total gyroscopic torque $\Omega = w_1 - w_2 + w_3 - w_4$, from Equation (12), we have

$$
\begin{bmatrix} \ddot{\psi} \\ \ddot{\theta} \\ \ddot{\phi} \end{bmatrix} = \begin{bmatrix} M_x - \frac{(J_z - J_y)\dot{\theta}\dot{\phi}}{J_x} \\ M_y - \frac{(J_x - J_z)\dot{\psi}\dot{\phi}}{J_y} \\ M_z - \frac{(J_y - J_x)\dot{\psi}\dot{\theta}}{J_z} \end{bmatrix} = \begin{bmatrix} -\frac{(J_z - J_y)\dot{\theta}\dot{\phi}}{J_x} + \frac{I_w\dot{\theta}\Omega}{J_x} + \frac{\sqrt{2}LU_2}{2J_x} \\ -\frac{(J_x - J_z)\dot{\psi}\dot{\phi}}{J_y} + \frac{I_w\dot{\psi}\Omega}{J_y} + \frac{\sqrt{2}LU_3}{2J_y} \\ -\frac{(J_y - J_x)\dot{\psi}\dot{\theta}}{J_z} + \frac{U_4}{J_z} \end{bmatrix}.
\tag{13}
$$

From Equations (9) and (13), the final dynamics model of the drone is obtained by neglecting the drag coefficient under slow flight conditions as follows:

$$
\begin{cases} \ddot{\psi} = -\frac{(J_z - J_y)\dot{\theta}\dot{\phi}}{J_x} + \frac{I_w\dot{\theta}\Omega}{J_x} + \frac{\sqrt{2}LU_2}{2J_x} \\ \ddot{\theta} = -\frac{(J_x - J_z)\dot{\psi}\dot{\phi}}{J_y} + \frac{I_w\dot{\psi}\Omega}{J_y} + \frac{\sqrt{2}LU_3}{2J_y} \\ \ddot{\phi} = -\frac{(J_y - J_x)\dot{\psi}\dot{\theta}}{J_z} + \frac{U_4}{J_z} \\ \ddot{x} = U_1\frac{c\psi s\theta c\varphi + s\varphi c\psi}{m} \\ \ddot{y} = U_1\frac{s\psi s\theta c\varphi - s\varphi c\psi}{m} \\ \ddot{z} = U_1\frac{c\theta c\varphi}{m} - g. \end{cases}
\tag{14}
$$

Ignoring the effect of rotor moment of inertia, the dynamical model can be linearized using the linear parameter varying method:

$$\begin{cases} \dot{X} = AX + BU \\ \dot{Y} = CX + DU \end{cases}, \tag{15}$$

where $A$, $B$, $C$, and $D$ are coefficient matrices; $X = \begin{bmatrix} \dot{x} & \dot{y} & \dot{z} & \dot{\varphi} & \dot{\theta} & \dot{\psi} & \varphi & \theta & \psi \end{bmatrix}^T$, $Y = \begin{bmatrix} \dot{z} & \dot{\varphi} & \dot{\theta} & \dot{\psi} \end{bmatrix}^T$, and $U = \begin{bmatrix} U_1 & U_2 & U_3 & U_4 \end{bmatrix}^T$. The system transfer function can be calculated from the AAID state space:

$$G_1(s) = C(SI - A)^{-1}B + D = \begin{bmatrix} \frac{1}{sm} & 0 & 0 & 0 \\ 0 & \frac{\sqrt{2}L}{2J_x s} & 0 & 0 \\ 0 & 0 & \frac{\sqrt{2}L}{2J_y s} & 0 \\ 0 & 0 & 0 & \frac{\sqrt{2}L}{2J_z s} \end{bmatrix}. \tag{16}$$

Ignoring the inductance coefficient of the motor and treating the model of the motor as a first-order inertial element, the transfer function of the motor torque is

$$G_2(s) = \begin{bmatrix} \frac{K}{\tau s+1} & 0 & 0 & 0 \\ 0 & \frac{K}{\tau s+1} & 0 & 0 \\ 0 & 0 & \frac{K}{\tau s+1} & 0 \\ 0 & 0 & 0 & \frac{K}{\tau s+1} \end{bmatrix}. \tag{17}$$

According to the actual measured parameters of AAID, the total mass of the machine is $m = 92$ g, the length of the quadrotor arm in the flight module is $h = 65$ mm, $J_x = 105895.96$ g·mm$^2$, $J_y = 108603.24$ g·mm$^2$, and $J_z = 198463.45$ g·mm$^2$. Considering the dynamic model of the hollow-cup motor as a first-order inertial element, $K$ in the transfer function $G_2(s)$ is the proportionality between the PWM control signal and the motor speed, and the final transfer function gives

$$G(s) = G_1(s)G_2(s) = \begin{bmatrix} \frac{0.082}{s(0.1s+1)} & 0 & 0 & 0 \\ 0 & \frac{3.10}{s(0.1s+1)} & 0 & 0 \\ 0 & 0 & \frac{3.10}{s(0.1s+1)} & 0 \\ 0 & 0 & 0 & \frac{37.5}{s(0.1s+1)} \end{bmatrix}. \tag{18}$$

## 3. Flight and Air–Land Transition Control Design

### 3.1. Flight Mode Optimal Control

3.1.1. Hovering Control

Optimal control is essentially a variational problem, and the most commonly used methods are the extreme value principle and dynamic programming. As shown in Figure 4, the optimal control problem is the determination of a control law for a given system so that the system can have the optimal value under the specified performance index under the given conditions. When the AAID is hovering, the system can be regarded as a linear system. According to the optimal control principle, the quadratic function of the state and control variables is selected as the optimal performance index, which enables the control of the AAID in the hovering state. It is assumed that the optimal control performance index function of the linear time-varying quadratic system is

$$J_{LQ} = \frac{1}{2} x^T(t_f) Q_0 x(t_f) + \frac{1}{2} \int_{t_0}^{\infty} x^T(t) Q_1(t) x(t) dt + \frac{1}{2} \int_{t_0}^{\infty} u^T(t) Q_2(t) u(t) dt, \tag{19}$$

where $Q_0$ denotes the $r$-dimensional symmetric positive definite constant matrix, $Q_1(t)$ denotes the $r$-dimensional symmetric positive semidefinite time-varying matrix, $Q_2(t)$

denotes the $m$-dimensional symmetric positive definite time-varying matrix, and $t_0$ denotes the initial time. Based on the established performance index, the optimal control law can be solved using the principle of minimum value. The $n$-dimensional vector $\lambda(t)$ is introduced to construct the Hamiltonian function:

$$H\begin{bmatrix} x & u & \lambda & t \end{bmatrix} = \frac{1}{2}[x^T(t)Q_1(t)x(t) + u^T(t)Q_2(t)u(t)] + \lambda^T[A(t)x(t) + B(t)u(t)]. \quad (20)$$

If the control performance index needs to be taken as an extreme value, then the Hamiltonian function takes a derivative of 0 concerning $u$, which gives

$$\frac{\partial H}{\partial u} = Q_2(t)u(t) + B^T(t)\lambda = 0. \quad (21)$$

The optimal control law $u^*$ is related to the vector $\lambda(t)$ as

$$u^* = -Q_2^{-1}(t)B^T(t)\lambda. \quad (22)$$

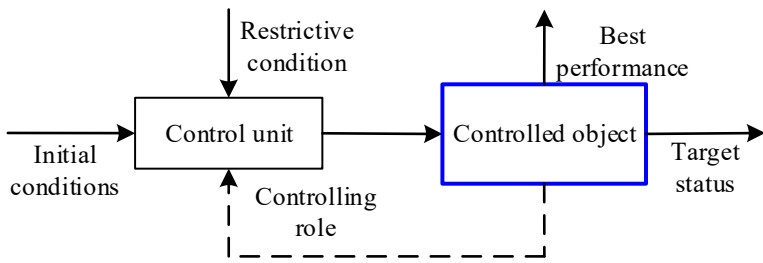

**Figure 4.** Optimal control principle.

If the relationship between the covariance vector $\lambda(t)$ and the state variable $x(t)$ is found, the state feedbacker can be obtained to achieve stable control of the system. The canonical equation is derived from the Hamiltonian function as follows:

$$\dot{x} = -\frac{\partial H}{\partial \lambda} = A(t)x(t) + B(t)u(t) = A(t)x(t) - B(t)Q_2^{-1}(t)B^T(t)\lambda; \quad (23)$$

$$\dot{\lambda} = -\frac{\partial H}{\partial x} = -Q_1(t)x(t) - A^T(t)\lambda. \quad (24)$$

From the above equation, $\lambda(t)$ is a linear function of $x(t)$. To obtain the state feedbacker, it is necessary to obtain the transformation matrix between $\lambda(t)$ and $x(t)$, assuming that

$$(t) = P(t)x(t); \quad (25)$$

$$u^* = -Q_2^{-1}(t)B^T(t)P(t)x(t) = -K(t)x(t). \quad (26)$$

Let $Q_2^{-1}(t)B^T(t)P(t) = K(t)$, and bring the resulting optimal control law $u^*$ into the original system, the state space expression of the closed-loop control system can thus be obtained:

$$\dot{x} = \left[A(t) - B(t)Q_2^{-1}(t)B^T(t)P(t)\right]x(t). \quad (27)$$

Substituting Equation (25) into Equations (23) and (24) and collating gives

$$\dot{P}(t) = -P(t)A(t) - A^T(t)P(t) + P(t)B(t)Q_2^{-1}(t)B^T(t)P(t) - Q_1(t). \quad (28)$$

The above equation is a Riccati matrix differential equation which is nonlinear, while $P(t)$ is symmetric [42]. Therefore, only $\frac{n(n+1)}{2}$ sets of first-order differential equations need to be solved to obtain the value of $P(t)$. According to the above analysis, the AAID at hovering is a linear constant system. $Q_2$ and $P$ in the Riccati matrix differential equation

are positive definite constant matrices, so $\dot{P}$ is 0 and $Q_1$ is a semi-positive definite constant array, which simplifies to give

$$PA + A^T P - PBQ_2^{-1}B^T P + Q_1 = 0. \tag{29}$$

### 3.1.2. Flight Control

Optimal control of the AAID flight posture requires augmenting the original matrix with a new state variable $x_{n+1}$:

$$\dot{x}_{n+1} = r_q - y = r_q - C(t)x, \tag{30}$$

where $r_q$ denotes the instruction, and the new state control function can be obtained from Equation (23):

$$\begin{bmatrix} \dot{x} \\ \dot{x}_{n+1} \end{bmatrix} = \begin{bmatrix} A & 0 \\ -C & 0 \end{bmatrix} \begin{bmatrix} x \\ x_{n+1} \end{bmatrix} + \begin{bmatrix} B \\ 0 \end{bmatrix} u + \begin{bmatrix} 0 \\ 1 \end{bmatrix} r_q. \tag{31}$$

The design state feedback control law is

$$\begin{bmatrix} -K_1 & -K_2 \end{bmatrix} \begin{bmatrix} x \\ x_{n+1} \end{bmatrix} = -K_1 x - K_2 x_{n+1}. \tag{32}$$

From Equation (26), the relationship between u and x can be obtained, and substituting Equation (31) gives

$$\begin{bmatrix} \dot{x} \\ \dot{x}_{n+1} \end{bmatrix} = \begin{bmatrix} A - BK_1 & -BK_2 \\ -C & 0 \end{bmatrix} \begin{bmatrix} x \\ x_{n+1} \end{bmatrix} + \begin{bmatrix} B \\ 1 \end{bmatrix} r_q. \tag{33}$$

The Laplace transform of Equation (33) gives

$$\begin{bmatrix} x(s) \\ x_{n+1}(s) \end{bmatrix} = \left( SI - \begin{bmatrix} A - BK_1 & -BK_2 \\ -C & 0 \end{bmatrix} \begin{bmatrix} x \\ x_{n+1} \end{bmatrix} \right)^{-1} \begin{bmatrix} 0 \\ r_q \end{bmatrix}. \tag{34}$$

When the reference input is a step signal from the final value theorem, we obtain

$$\lim_{t \to \infty} \begin{bmatrix} x(t) \\ x_{n+1}(t) \end{bmatrix} = \lim_{s \to 0} \begin{bmatrix} x(s) \\ x_{n+1}(s) \end{bmatrix} = -\begin{bmatrix} A - BK_1 & -BK_2 \\ -C & 0 \end{bmatrix}^{-1} \begin{bmatrix} 0 \\ r_q \end{bmatrix}. \tag{35}$$

It can be seen that $x(t)$, $x_{n+1}(t)$ tends to a constant value, and $\dot{x}(t)$, $\dot{x}_{n+1}(t)$ tends to zero so that the output can be tracked.

### 3.1.3. Simulation Analysis

The optimal control algorithm of the AAID is simulated by taking the hovering mode as an example. Using the state feedbacker of Equations (23) and (26), whether the control of each direction of the AAID can return to the initial state under the state of external disturbance can be observed. During the simulation process, certain errors are applied to the pitch, roll, and yaw angles of the AAID, respectively, and the response time results are observed as shown in Figure 5. It can be found that when the interference is applied simultaneously to each direction of the AAID for hovering flight, the AAID can quickly backtrack to the target position even if the errors are applied to the pitch, roll, and yaw angles. There is a decreasing oscillation of the flight position error in the first 5 s of the simulation, and then it converges to the target position, which can achieve stable hovering control of the AAID.

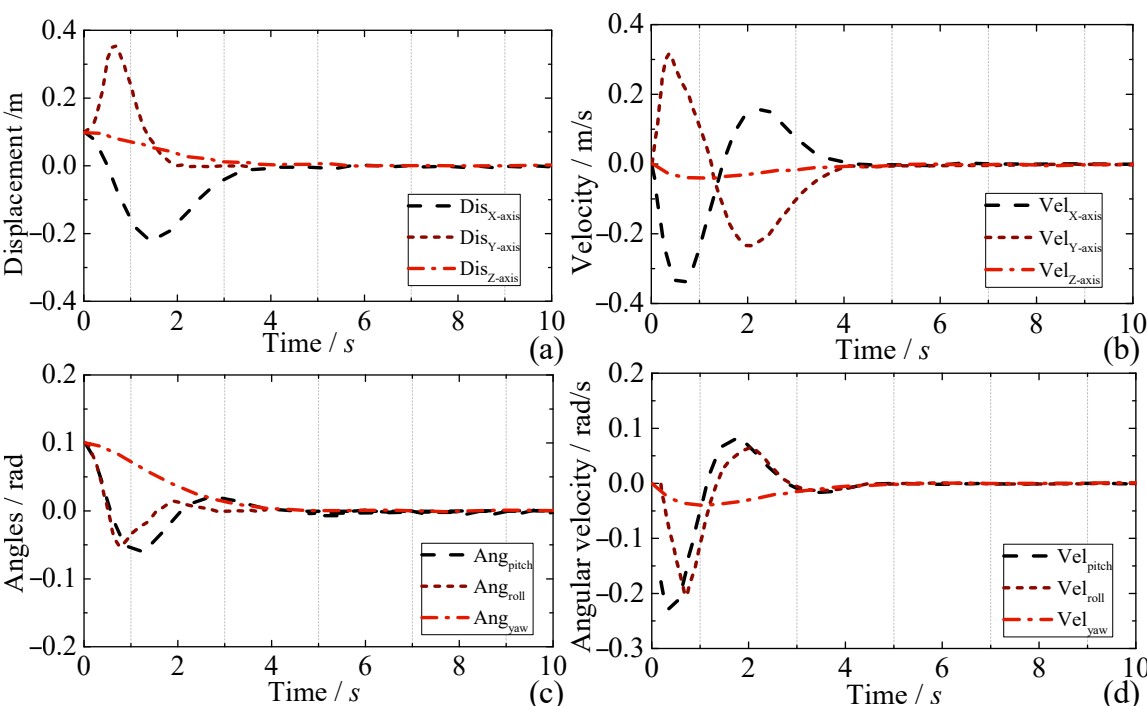

**Figure 5.** Dynamic response of AAID in hover mode: (**a**,**b**) *X*-axis, *Y*-axis, and *Z*-axis direction displacement and velocity; (**c**,**d**) pitch, roll, yaw direction angle, and angular velocity.

### 3.2. Flying–Crawling Transition Control

The AAID needs to be capable of stable flight and hovering, as well as flying–crawling transition in appropriate situations. As shown in Figure 6, when an obstacle is found in front of the AAID that cannot be crawled over, the AAID will switch to a flight control mode to achieve obstacle crossing by rotor flight. When encountering a flat road surface, it is necessary to go from flight mode to crawl mode to save energy. The AAID flying–crawling transition control flow is shown in Figure 7. First, the AAID senses whether there is an obstacle in front of it according to the infrared photoelectric sensor; when an obstacle is detected, the infrared photoelectric sensor outputs pulses, which are inputted to the MCU for processing, and then the motor drive module is controlled. At this point, the AAID will stop crawling and sound an alarm. When a flight command is received, the AAID switches to the flight model and crosses the obstacle. Since the AAID flight mode consumes more energy, it needs to be switched to crawl mode when the environment is suitable. The AAID mode switching mainly relies on the barometer to obtain the height. When the AAID reaches a suitable altitude, it starts to decelerate and land, and when the *Z*-axis speed of the AAID is detected to be zero, it switches to the crawling model and continues to move forward. Repeating the above process can achieve amphibious movement of the AAID.

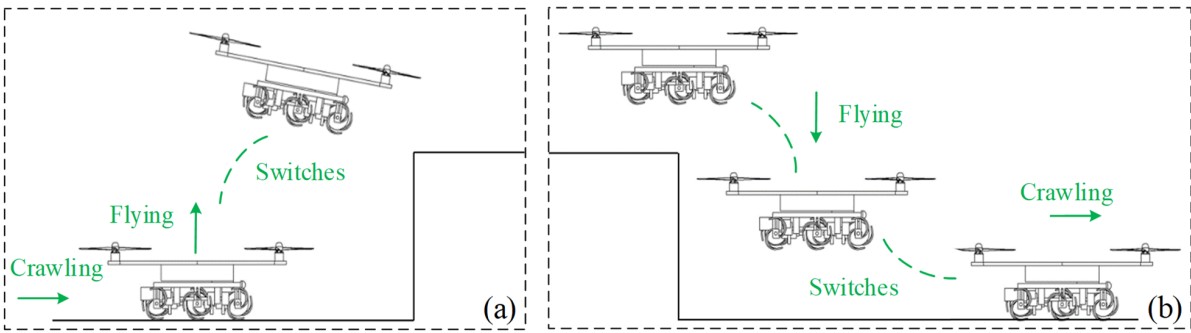

**Figure 6.** Fly–crawl conversion control process: (**a**) the AAID detects an obstacle and switches to flight mode; (**b**) the AAID detects a flat area and switches to crawl mode.

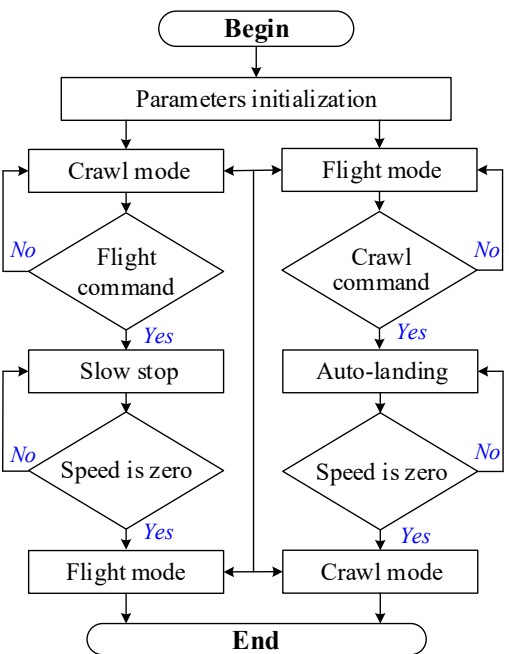

**Figure 7.** Flying–crawling transition control strategy.

## 4. Prototype Experiments

### 4.1. Flying–Crawling Transition Control

To meet the AAID miniature and lightweight design requirements, the rack is made of PCBs with integrated control circuits, with a size of 100 mm × 100 mm × 2mm and a weight of 16.75 g. The control board adopts a four-layer structure; the top and bottom layers have the layout of electronic components, and the power supply layer carries out block power supply to meet the needs of different electronic components, as shown in Figure 8a. The assembled AAID is shown in Figure 8b. The AAID control system is based on a dual MCU model, the main chip is an STM32F411 with a 32 bit Cortex-M4 core with up to 256-to-512 KB of Flash memory and up to 128 KB of SRAM. The role of the main control MCU includes sensor data reading, data calculation, power control, wireless communication, etc. As the brain of the AAID, it has a decisive role in stable flight in the air. Besides the main MCU, the AAID has a secondary MCU, the wireless chip NRF51822. Besides wireless communication, it also undertakes power management, power amplification, and other tasks. In the power-off state, the secondary MCU (wireless chip NRF51822) runs on standby and the main MCU (STM32F4111) is in the power-off state. When the system is powered up, the secondary MCU is woken up, and then the main MCU enters firmware mode. The workflow of the AAID control system designed in this paper is as follows: power-on self-test, initialization of the system, receiving remote control commands, setting flight parameters, acquiring flight attitude, data fusion, solving attitude information, calculating control quantities, outputting control signals, and finally completing motion control.

### 4.2. Crawling Experiment

To verify the crawling and obstacle-crossing ability of the wheel-legged crawling system, the experimental platform with different slopes is built by using wooden boards with uniform roughness.

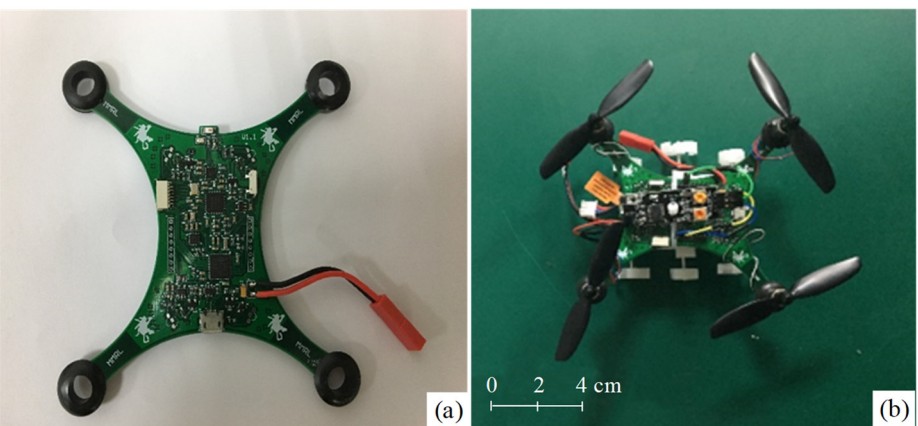

**Figure 8.** AAID prototype: (**a**) quadrotor airframe control system; (**b**) AAID fly–crawl integrated prototype.

Eight types of rough surfaces, 10-mesh, 12-mesh, 14-mesh, 16-mesh, 18-mesh, 20-mesh, 25-mesh, and 35-mesh, are fabricated by choosing quartzite with different particle diameters as the substrate. The crawling data are recorded three times for each slope during the experiment, and the crawling ability is tested at a constant speed on the surface with different roughness. Figure 9a–d shows the results of the AAID crawling experiment at 14-mesh, 20° slope. The experimental video can be seen in the Supplementary Materials. The AAID can easily crawl over a 20° slope on all types of surfaces with different levels of roughness. When the angle is increased to 25°, the crawler system slips severely, as shown in Figure 9e–h, and overturns after a short distance. When the angle is in the range of 21–24°, the risk of overturning is very high, although successful passage is occasionally possible. The main reason for this is that the crawler system will have a slight jump when crossing the obstacle, resulting in a relatively large overturning torque. As shown in Figure 9e–h, when the angle is small ($2F \ll G\cos\theta$), the reaction force generated is not enough to cause the overturning of the crawling system. When the angle slowly increases, and the vertical component of gravity concerning the inclined plane is not sufficient to counteract the tipping torque generated by $F$, the front side of the fuselage will jump more significantly. Crawling on the inclined surface will cause overturning, resulting in crawling failure. After several tests, the maximum crawling height for stable crawling without overturning is 20°.

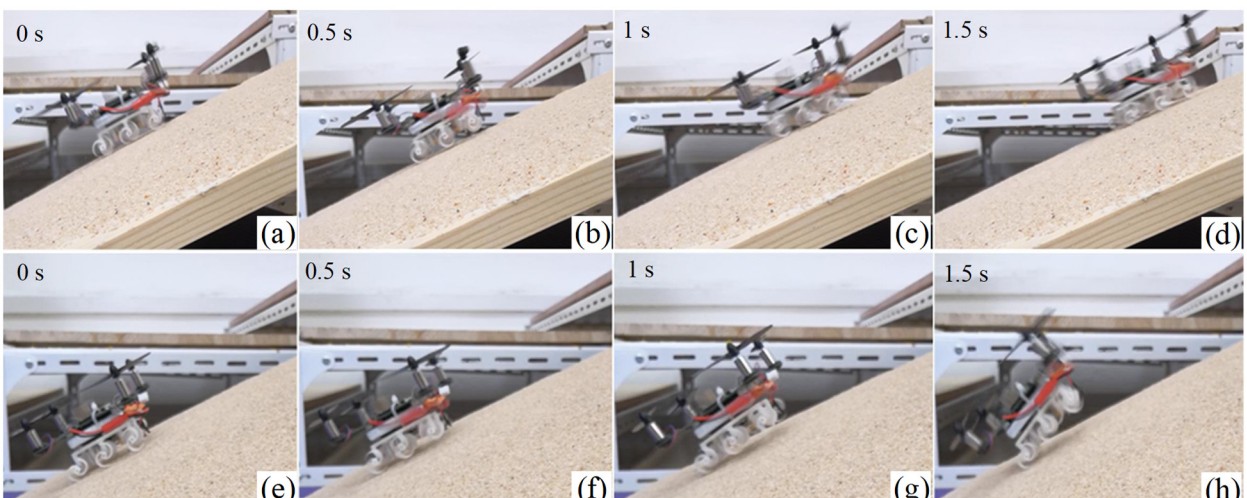

**Figure 9.** (**a**–**d**) The 14-mesh 20° crawl experiment; (**e**–**h**) the 14-mesh 25° crawl experiment.

### 4.3. Hovering Experiment

The AAID needs to have the ability to hover stably in the air; for this reason, we carried out the AAID hovering flight experiment, as shown in Figure 10. During the experiment, only the control signal of the Z-axis of the AAID is set, and the hovering effects under different control algorithms are analyzed according to the stability of the roll angle, pitch angle, and yaw angle of the AAID during hovering, and the results are shown in Figure 11. When LQ is used to optimally control the roll angle, pitch angle, and yaw angle during hovering, the changes are within $\pm 2°$, and the whole error curve changes very smoothly with high stability. When the serial PID control is used, the roll angle, pitch angle, and yaw angle change within $\pm 4°$ during hovering. When fuzzy PID control is used, the variation of attitude angle in all directions is also within $\pm 2°$, but the stability of fuzzy PID deteriorates and the shaking is very obvious in the hovering state. Therefore, the posture control effect under LQ control is better than that of serial PID and fuzzy PID, which proves the effectiveness of the LQ control algorithm.

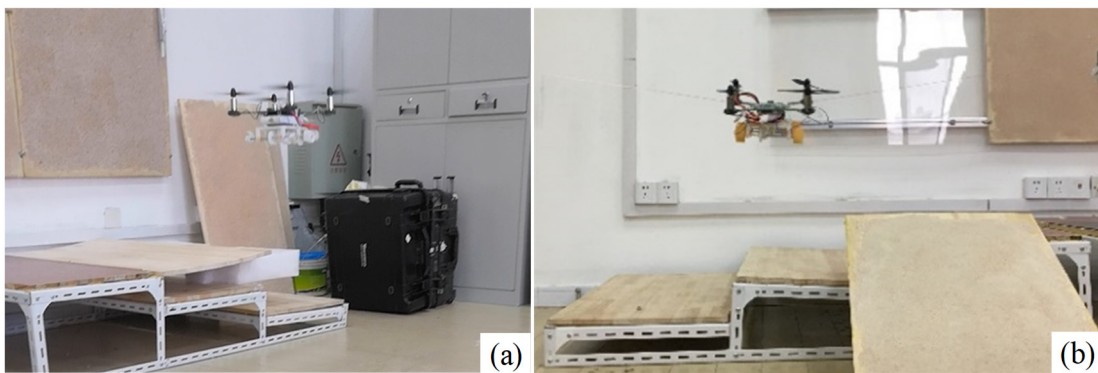

**Figure 10.** AAID hovering flight experiment. (**a**,**b**) AAID hovering side view and main view.

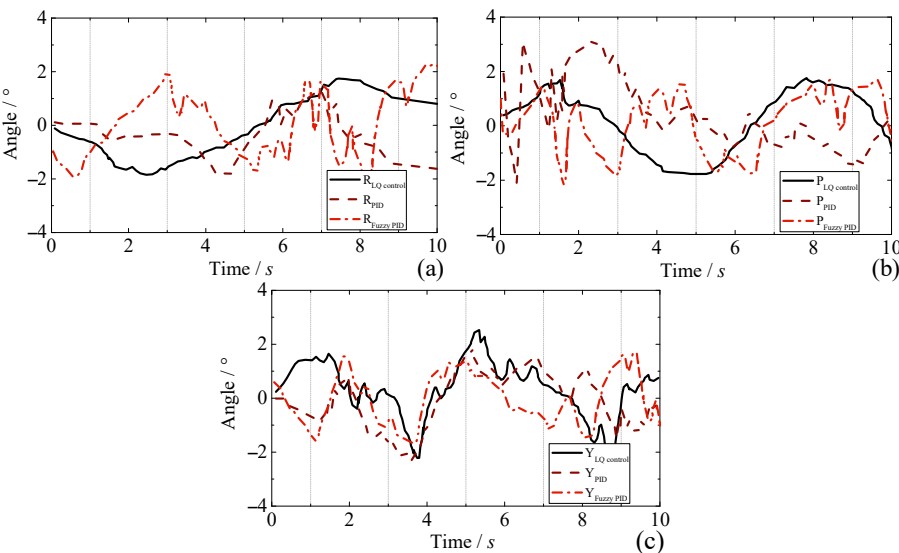

**Figure 11.** Dynamic response of hovering flight under different control laws. (**a–c**) Response curves for roll angle, pitch angle, and yaw angle.

### 4.4. Fly–Crawling Transition Experiment

For the AAID fly–crawl conversion experiment, an experimental platform was built indoors, as shown in Figure 12a. When the AAID performs ground crawling, the infrared sensor detects whether there is an obstacle in front of it. If an obstacle appears, the crawling is stopped, and the alarm is issued. After waiting for the flight command to cross the obstacle, it will slowly land on the ground to complete the fly–crawl transition experiment.

Figure [12]b–d show the process of the AAID encountering and crossing obstacles. The distance from the obstacle when the AAID stops is recorded through several experiments, as shown in Table. 3. It can be seen that the average obstacle avoidance distance of the AAID is at 128.8 mm, and this distance is the distance of the foremost motor of the AAID from the obstacle. Considering the size of the AAID propeller, half of the distance of the propeller (37.5 mm) needs to be reserved. Then, when the average distance of AAID from the obstacle is 91.3 mm, it can realize the automatic shutdown of the crawler system and send out an alarm. To verify the landing stability of the AAID, the experiments of AAID stable landing are counted, as shown in Table [3], and the success rate of AAID stable landing is 77%. The main reason for the landing failure is that the AAID uses a barometer to obtain the height data, which has a certain delay. In addition, the AAID landed stably according to the speed of the Z-axis; due to the rigidity of the AAID, the wheel-leg structure of the ground crawling system will produce a rebound movement when it comes into contact with the ground, so that the Z-axis speed of the AAID was not reduced to 0 when landing, failing stable landing.

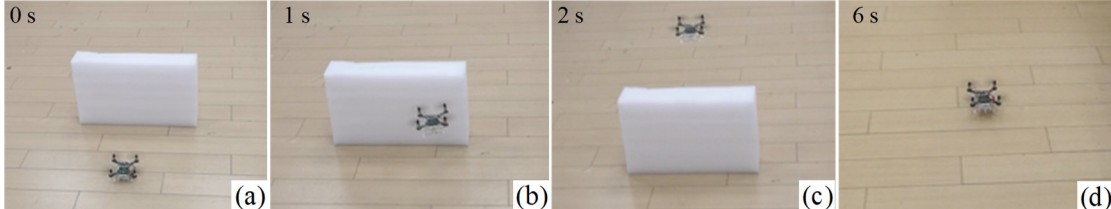

**Figure 12.** AAID obstacle avoidance experiment process. (**a**–**d**) Timing diagram of obstacle avoidance at different times for AAID.

**Table 3.** AAID autonomous obstacle avoidance success rate statistics.

| No. | 1 | 2 | 3 | 4 | 5 | 6 | 7 | 8 | 9 |
|-----|-----|-----|-----|-----|-----|-----|-----|-----|-----|
| Distance/mm | 128 | 135 | 120 | 139 | 119 | 130 | 128 | 125 | 133 |
| Success rate | Y | Y | N | Y | Y | N | Y | Y | Y |

## 5. Conclusions

Remote handling is one of the challenges that must be solved to lead a magnetic confinement fusion device to commercial operation. In this paper, a new design method for a miniature air–land amphibious inspection drone (AAID) is proposed for the fusion reactor discharge gap observation requirements. Through the flying and crawling amphibious function, the AAID can realize the narrow maintenance channel crawling transportation and fusion reactor internal flight observation to meet the needs of discharge gap inspection and transportation. To realize miniaturization and energy saving, the ground platform adopts a cockroach-like wheel-legged system to enhance the obstacle-crossing ability. The flying platform adopts the rotor structure with a PCB board integrating the rack and control system to reduce the weight of the whole aircraft. The integrated AAID weighs only 92 g, and the overall dimensions are less than 100 mm × 100 mm × 60 mm, which can meet the transportation requirements for accessing the maintenance window of the fusion reactor. The AAID has the characteristics of nonlinearity, strong coupling, and multi-variability. Based on the dynamic model and the optimal control method, the control strategies under flight mode, hover mode, and fly–crawl transition are designed, respectively. The simulation results show that the proposed linear quadratic optimal control method can make the AAID reach the desired position faster with a stable posture. Finally, the prototype of the AAID is constructed, and the crawling, hovering, and fly–crawling transition control experiments are carried out, respectively. In the crawling stage, we implemented a variety of different roughnesses of the crawling surface. The crawling platform can easily crawl through the 20° slope; however, the larger the diameter of the surface substrate particles, the weaker the crawling ability of the crawling platform. When

the angle is increased to 25°, the platform slips severely and overturns after crawling for a certain distance. In the hovering phase, we compare and analyze the superiority of serial PID control, fuzzy PID control, and the linear quadratic optimal control proposed in this paper. All the selected control methods can make the AAID maintain stable hovering in a small range. The roll angle, pitch angle, and yaw angle deviation of the AAID with linear quadratic optimal control is less than 2°, with the smallest error and the highest stability proving the effectiveness of the proposed method. In the fly–crawl transition phase, we conducted several transition experiments. The success rate of stabilized landing of the AAID has only reached 77%. The main reason for this is that the AAID landed based on the velocity in the Z-axis direction. The rigid contact produces an upward rebound velocity, which causes the failure of a stabilized landing. The success rate can be improved by compensating with motion control algorithms. Overall, the amphibious inspection AAID structural design and dynamic control strategy are effective and can provide a reference for fusion reactor inspection drone system development.

In the future, we will further develop the AAID system to improve the landing success rate through motion planning and wheel-leg optimization. At the same time, we will develop a binocular stereovision system to further improve the function of inspection drones.

**Supplementary Materials:** The following supporting information can be downloaded at: https://www.mdpi.com/article/10.3390/drones8050190/s1.

**Author Contributions:** Conceptualization, G.Q.; Data curation, G.Q. and W.H.; Formal analysis, Y.X., Q.Q., L.Z. and H.H.; Funding acquisition, G.Q. and Y.C.; Investigation, C.Z. and A.J.; Methodology, W.H. and Q.Q.; Project administration, Y.C.; Resources, L.Z. and Q.Q.; Supervision, C.Z., D.Z. and A.J.; Validation, G.Q.; Visualization, G.Q. and W.H.; Writing—original draft, G.Q. and W.H.; Writing—review and editing, Y.X., H.H. and D.Z. All authors have read and agreed to the published version of the manuscript.

**Funding:** This work was supported by the National Natural Science Foundation of China (12305251), the Postdoctoral Fellowship Program of CPSF (grant nos. GZB20230770, PF230101018), and the Comprehensive Research Facility for Fusion Technology Program of China (grant nos. 2018-000052-73-01-001228).

**Institutional Review Board Statement:** Not applicable.

**Informed Consent Statement:** Not applicable.

**Data Availability Statement:** The data and code used to support the findings of this study are available from the corresponding author upon request (gdqin@ipp.ac.cn).

**Conflicts of Interest:** The authors declare no conflicts of interest.

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
