# Peer review of "Design and Development of an Air–Land Amphibious Inspection Drone for Fusion Reactor"

_drones, doi:10.3390/drones8050190_

Round 1

Reviewer 1 Report

Comments and Suggestions for Authors

The paper proposes a design method for a miniature air-land amphibious inspection drone to be used in the compact fusion reactor discharge gap observation mission.

To fit a mission profile (that it is not incorporated into the paper) authors realized a level of miniaturization that is not, as said, related to a defined target specs, adopting the bionic cockroach wheel-legged system to improve (they wrote) the obstacle crossing ability (again; no targets are included into the paper). Design method and modeling are, anyway, very good and appreciated. In brief just a specific mission profile set of parameters, where to clarify the state of the art of the application/environment (compact fusion reactor discharge gap observation mission), and the relevant improved target spec are missing. A complete evaluation cannot be so performed.

Author Response

Thanks for the experts' comments, we have responded to each question and uploaded them to the attachment please check. If there are any further questions please feel free to feedback we will revise them as soon as possible.

Reviewer 2 Report

Comments and Suggestions for Authors

The equations 11 to 14 have to be rewritten, they are not clear or coherent with the following ones. some of the symbols are not clearly described, authors should consider a nomenclature section to allow a better reading for the readers.

Figures 2 and 3 do not help the reader to understand the definition of the main angles and the Ff force is not clearly identifiable from pictures.

Comments on the Quality of English Language

Sufficient

Author Response

(The authors gave the same response as above.)

Reviewer 3 Report

Comments and Suggestions for Authors

The article is well structured with a clear focus, goal and methodology. Also, the authors gave the possibility of application in practice. I believe that the paper can be accepted.

The manuscript is well-structured,  clear and relevant for the field.

Author Response

Thanks for your comments. In the future, we will continue to strive to write more excellent manuscripts for continued publication in this journal.

Reviewer 4 Report

Comments and Suggestions for Authors

This paper presents the development of a miniature air-land amphibious drone designed for inspecting fusion reactors. It details the drone's bionic design, dynamic control strategies, and testing outcomes, highlighting its efficient transition between crawling and flying modes. However, some issues need to be addressed:

1.      The introduction section requires a clearer articulation of the novelty and contributions of the AAID design compared to existing technologies, particularly for nuclear reactor inspection applications. Why is AAID superior to the alternatives presented in the literature?

2.      The introduction also needs a clearer explanation of why the drone requires a hybrid mode design. It's not apparent why a conventional drone cannot accomplish the inspection mission.

3.      The drone design does not include any inspection payload. The authors need to explain how they manage to conduct inspections without any sensing payload. What is the payload limitation—1 kg, 2 kg, or more? What are the impacts of adding payloads?

4.      The experiments lack comprehensive details about the environmental conditions during tests, which are crucial for the novelty of this paper. For instance, the details of surface roughness and hardness are missing. Additionally, did you test the drone on metallic surfaces? According to Figure 1, it seems that the surfaces inside a nuclear reactor are not made of wood, so why were tests conducted on a wooden surface? Moreover, the authors mentioned using wireless communication; is it possible to receive a wireless signal inside a reactor? What about potential radio interference? These aspects require further experiments and should be included in the experimental section.

5.      The hovering experiment results show that the errors are within 4 degrees. What are the requirements to complete inspection missions? Are you planning to use a visual camera, infrared sensor, or something else? What are the limitations of these inspection payloads in achieving meaningful inspections? In other words, an error margin of 4 degrees might not be sufficient. Please discuss these aspects

6.      Were experiments conducted at 21, 22, 23, and 24 degrees? If not, these should be performed. Jumping from a 20-degree to a 25-degree slope creates a significant gap, making it difficult to conclude the system's effectiveness at 20 degrees based on its failure at 25 degrees. 

Author Response

(The authors gave the same response as above.)

Round 2

Reviewer 4 Report

Comments and Suggestions for Authors

Figure 3 caption is incorrect. CFETR and COMR never mentioned before.

Author Response

Thanks for your comments. We apologise for the misuse of the caption of figure 3 in the manuscript due to our error. Based on the expert's comments we have revised the caption of figure 3 and carefully checked the caption of each image. Please check. (Please check p. 5 lines 165-166)
